# Using a Combination of Novel Research Tools to Understand Social Interaction in the *Drosophila melanogaster* Model for Fragile X Syndrome

**DOI:** 10.3390/biology13060432

**Published:** 2024-06-12

**Authors:** Maja Stojkovic, Milan Petrovic, Maria Capovilla, Sara Milojevic, Vedrana Makevic, Dejan B. Budimirovic, Louise Corscadden, Shuhan He, Dragana Protic

**Affiliations:** 1Department of Pharmacology, Clinical Pharmacology and Toxicology, Faculty of Medicine, University of Belgrade, 11000 Belgrade, Serbia; maja.stojkovic@med.bg.ac.rs (M.S.); sara.milojevic@med.bg.ac.rs (S.M.); 2Department of Informatics, University of Rijeka, 51000 Rijeka, Croatia; milan.petrovic@uniri.hr; 3Center for Artificial Intelligence and Cybersecurity, University of Rijeka, 51000 Rijeka, Croatia; 4UMR7275 CNRS-UCA, Institut de Pharmacologie Moléculaire et Cellulaire, 06560 Valbonne Sophia Antipolis, France; maria.capovilla@ipmc.cnrs.fr; 5Department of Pathophysiology, Faculty of Medicine, University of Belgrade, 11000 Belgrade, Serbia; vedrana.parlic@med.bg.ac.rs; 6Department of Psychiatry, Fragile X Clinic, Kennedy Krieger Institute, Baltimore, MD 21205, USA; budimirovic@kennedykrieger.org; 7Department of Psychiatry & Behavioral Sciences-Child Psychiatry, School of Medicine, Johns Hopkins University, Baltimore, MD 21205, USA; 8Maze Engineers, Skokie, IL 60077, USA; louise@mazeengineers.com; 9Lab of Computer Science, Department of Internal Medicine, Harvard Medical School, Boston, MA 02115, USA; she@mgh.harvard.edu; 10Fragile X Clinic, Special Hospital for Cerebral Palsy and Developmental Neurology, 11000 Belgrade, Serbia

**Keywords:** *Drosophila melanogaster* model of fragile X syndrome, *FMR1* gene, fragile X syndrome, social anxiety, social interaction, social network analysis

## Abstract

**Simple Summary:**

Fragile X syndrome (FXS) is a genetic disorder causing intellectual disability and autism spectrum disorder and it is the result of a full mutation in the Fragile X Messenger Ribonucleoprotein 1 (*FMR1*) gene. Individuals with FXS are presented with various behavioral challenges, including social anxiety and difficulties in social interactions (SI). Scientists have used animals like fruit flies to study FXS. In these models, mutations in the gene similar to the human *FMR1* have helped researchers gain insights into the condition. The current study utilized the fruit fly’ model of FXS to understand SI patterns, and their differences compared to flies without the mutation. The results showed that mutant flies exhibited reduced activity, struggled to form connections and had difficulty effectively sharing information. These findings suggest distinct social patterns in mutant flies, shedding light on the social challenges associated with FXS. Importantly, this study demonstrates how innovative research tools can lead to a better understanding of social challenges associated with FXS and identify potential treatments.

**Abstract:**

Fragile X syndrome (FXS), the most common monogenic cause of inherited intellectual disability and autism spectrum disorder, is caused by a full mutation (>200 CGG repeats) in the Fragile X Messenger Ribonucleoprotein 1 (*FMR1*) gene. Individuals with FXS experience various challenges related to social interaction (SI). Animal models, such as the *Drosophila melanogaster* model for FXS where the only ortholog of human *FMR1* (*dFMR1*) is mutated, have played a crucial role in the understanding of FXS. The aim of this study was to investigate SI in the *dFMR1^B55^* mutants (the groups of flies of both sexes simultaneously) using the novel Drosophila Shallow Chamber and a Python data processing pipeline based on social network analysis (SNA). In comparison with wild-type flies (*w^1118^*), SNA analysis in *dFMR1^B55^* mutants revealed hypoactivity, fewer connections in their networks, longer interaction duration, a lower ability to transmit information efficiently, fewer alternative pathways for information transmission, a higher variability in the number of interactions they achieved, and flies tended to stay near the boundaries of the testing chamber. These observed alterations indicate the presence of characteristic strain-dependent social networks in *dFMR1^B55^* flies, commonly referred to as the group phenotype. Finally, combining novel research tools is a valuable method for SI research in fruit flies.

## 1. Introduction

Social interactions (SIs) include various interactions among individuals and play a fundamental role in their lives. These interactions can be complex and diverse, often involving a range of behaviors, such as mating, aggression, dominance, vocalizations, and body language [1]. Over time, as individuals within the group interact with each other, specific patterns and dynamics emerge. Disruption in normal social behavior and SI is common in various human diseases and conditions, such as neurodevelopmental disorders [2]. An example of a neurodevelopmental disorder characterized by impaired SI is fragile X syndrome (FXS).

FXS is caused by a full mutation (>200 CGG trinucleotide repeats) in the Fragile X Messenger Ribonucleoprotein 1 (*FMR1*) gene and is the most common monogenic cause of inherited intellectual disability and autism spectrum disorder. The *FMR1* gene encodes the *FMR1* protein (FMRP) which is responsible for the translation of messenger RNAs (mRNAs), RNA stability, sub-cellular transport, regulation of ion channels activity, synaptic development and plasticity, and has many other roles [3,4]. Among other symptoms, individuals with FXS experience a wide range of challenges related to SI including challenges in maintaining eye contact, shyness, social anxiety, social withdrawal, and social avoidance [5,6]. Impairment of SI often causes daily struggles that significantly impact the ability of FXS individuals to engage in typical daily activities [7].

Animal models have played a crucial role in advancing the understanding of FXS. Commonly used animal models include *Fmr1* knock-out (KO) mice, *Fmr1* KO zebrafish, and the *Drosophila melanogaster* (*D. melanogaster*) model of FXS, where the only ortholog of human *FMR1* (*Fmr1*, FlyBase ID: FBgn0028734, herein *dFMR1*) is mutated [8,9,10,11,12]. *D. melanogaster*, commonly known as the fruit fly, has a well-characterized nervous system and genetic manipulations can be performed to mimic the genetic mutations associated with FXS. *D. melanogaster* represents a valuable model for understanding the molecular and neurological aspects of the syndrome [10,11]. It is well known that phenotypes in *dFMR1* mutants closely resemble the phenotypes in individuals with FXS. For example, *dFMR1* mutants display arrhythmic circadian rhythm, abnormal locomotor activity and learning and memory deficits, similar to the symptoms of FXS [13,14,15]. Based on FlyBase data, dFMRP, the protein that is encoded by *dFMR1*, participates in over 50 biological processes, affecting both neuronal and non-neuronal functions in *Drosophila*. Its most significant contribution is to synaptic plasticity. Additionally, dFMRP is crucial in aging, apoptosis, phagocytosis, and numerous other processes [10]. Despite *D. melanogaster* being an excellent model for studying FXS, there is currently limited data on SI in this model. On the other hand, some studies described SI in other fruit fly models. For example, SI impairments were described in *orco*, *lush*, and *or65a* mutants [16,17,18]. Exploring and collecting more information on SI in the *D. melanogaster* model of FXS could significantly enhance FXS research and contribute to the preclinical evaluation of drug effects in this condition.

The aim of this study was to investigate, analyze and describe SI in the FXS model of *D. melanogaster* using the novel chamber and a Python data (Python Software Foundation, Beaverton, OR, USA) processing pipeline based on social network analysis (SNA).

## 2. Materials and Methods

### 2.1. Flies

The *dFMR1^B55^* allele was generated by imprecise excision of the *EP(3)3422* element that caused a 2.5 kb deletion of *dFMR1* genomic DNA including the ATG and the first 59 codons [19]. Thus, B55 is a protein null allele. *dFMR1^B55^* flies are homozygous viable and fertile. The wild-type *w^1118^* flies were used as a control group in all experiments.

*D. melanogaster* stocks were reared on standard cornmeal/molasses/agar medium at 25 °C and at a relative humidity of 60% under a 12 h light/12 h dark cycle [20]. At the time of eclosion, the flies were collected under light anesthesia (CO_2_) and grouped by age. Each vial contained 30 flies (15 of each sex), which were kept in an incubator until they reached the age of 3–5 days, after which they were used in the experiments [20]. On the day of the experiment, the flies were gently transferred from the vials to the arena using an aspirator and left for 15 min to habituate [21]. To ensure the most accurate results and minimize performance variability linked to circadian rhythm, all experiments were conducted in the afternoon, between ZT5 and ZT9 [20]. 

### 2.2. Drosophila Shallow Chamber

The Drosophila Shallow Chamber (Maze Engineers, Skokie, IL, USA) is designed to restrict *D. melanogaster* flies to a shallow space to create a monolayer of individuals for behavioral analysis. The chamber is cylindrical and composed of clear acrylic with a diameter of 13 cm and a 3.5 mm high glass ceiling coated with silicone paint. The chamber is surrounded by translucent checkered black and white paper to stimulate the movement of flies within the chamber. The walls of the chamber are at an 11-degree angle downwards, preventing the subjects from gathering on the ceiling of the chamber. Backlighting underneath the chamber features a 12 × 12 inch fluorescent light array of 850 nm LEDs. 

### 2.3. Experimental Design

Groups of 30 adult flies, comprising 15 of each sex per vial, were housed together in the same vial within the incubator from eclosion until they reached the age of 3–5 days, as detailed in Section 2.1. On the day of the experiment, using an aspirator, these groups were transferred to the Drosophila Shallow Chamber, described in Section 2.2, where they were video recorded. Movies were recorded for 15 min at 60 frames per second [18]. In total, 15 *dFMR1^B55^* and 15 *w^1118^* separate groups of flies were recorded and tested. The chamber and the cover were cleaned with 75% ethanol between each experiment to remove potential residues [22]. The schematic representation of the experimental design is shown in Figure 1 and explained in Appendix A.

### 2.4. Data Analysis

#### 2.4.1. Fly Tracking 

Fly tracking was done using the open source software Caltech FlyTracker 1.1.2, developed using MATLAB v. R2023a. FlyTracker is a reliable tracking tool that provides multiple fly position and orientation data in each video frame and maintains their identities [23]. It outputs trajectories and features such as velocity, facing angles to other flies and wing angles [22,24]. Manual verification of identity swapping in tracking was performed in a randomly selected sample of 300 frames and no loss or swap of identities was recorded.

#### 2.4.2. Construction of Social Interaction Networks (SINs) and Social Network Analysis (SNA)

A Python data processing pipeline was developed for the dual purposes of calculating activity levels, encompassing total distance walked and average velocity, and for the construction and analysis of Social Interaction Networks (SINs) [25].

SINs are depicted as weighted graphs G = (V, E), comprising two distinct sets. V nodes (vertices) represent individual flies, while E consists of links (edges) with associated weights, quantifying interactions between flies. We have introduced two weight factors: (i) the count of interactions and (ii) the total interaction duration. Specifically, every interaction between files is represented by a single link, where the weight reflects both the count of different interactions and the cumulative interaction duration. This approach determines the weights of links within the network for each fly, accounting for the number of interactions and the total interaction duration during the 15 min video [25]. Due to the large number of interactions, Social Network Analysis (SNA) was performed using a temporal network analysis in which each minute of the experiment is represented by a separate network snapshot [26]. Measurements were then taken across each representation of networks, which were summarized in the distribution for each group.

The criteria for SI were: (i) the distance between flies is within two-and-a-half body lengths, (ii) the angle between flies is less than 160°, and (iii) previous conditions are met for a duration longer than 0.6 s [24].

SINs of the two investigated groups were analyzed using the following measures:

*Total edges* represent the overall number of edges (interactions) among nodes (flies). The measure represents the total number of interactions during the experiment:

*The Average degree* refers to the number of edges (interactions) connected to a single node (fly). It measures the average number of interactions per fly in the network [25]. 

*Average strength*: In networks with weighted edges, strength is the sum of all edge (interaction) weights connected to the node (fly). “In-strength” refers to the sum of all edge (interaction) weights a node (fly) receives, and “out-strength” refers to the sum of all edge (interaction) weights a node (fly) outputs [24]. 

*Network density*: The proportion of how many connections in a network exist compared to the number of theoretically possible connections. It indicates how closely interconnected the nodes (flies) are within a network [18]. 

*Global efficiency*: Distinguishes whether the overall network has shorter or longer paths between nodes (flies) and measures how efficiently information can be transferred across a network [24].

*Degree heterogeneity*: Measures the diversity in the node (fly) degrees and the diversity in the network’s structure [25].

*Degree assortativity*: Measures whether nodes (flies) with similar degrees are more likely to interact with each other or if there is a preference for nodes (flies) with a different degree. A positive assortativity indicates that nodes (flies) with similar degrees are more likely to interact. In contrast a negative assortativity indicates that nodes (flies) with different degree are more likely to interact [27].

*Transitivity*: Refers to the tendency for nodes (flies) to form clusters or triangles within the network. It assesses the likelihood that if two nodes (flies) are connected to the same node (fly), they are also connected [28].

*Average clustering coefficient*: Quantifies the degree to which nodes (flies) in a network tend to cluster together. It assesses how interconnected nodes are to one another [26].

*Average betweenness centrality*: Quantifies the importance of a node (fly) within a network. In other words, a fly with high betweenness centrality bridges information flow between different network parts [18].

*Average closeness centrality:* Measures how close a fly (node) is to other flies (nodes) in the network. High closeness centrality means that more flies are relatively close to other flies in the network [26].

*Modularity*: A measure used to assess the degree of community structure or clustering within a network. It evaluates how well a network can be divided into distinct groups of nodes (flies) with more interactions within the same group than with nodes outside the group [18].

#### 2.4.3. Localization of Social Interactions

To determine the frequency of interactions in relation to the localization within the chamber, a heatmaps were generated using the Matplotlib library in Python [25].

#### 2.4.4. Statistical Analysis

Statistical analysis was performed using GraphPad Prism 8.0 (GraphPad Software, Inc., San Diego, CA, USA). Data were presented as the mean ± standard deviation (S.D.) of the mean. The differences between the groups were analyzed using an unpaired Student’s *t*-test. *p* < 0.05 was considered statistically significant.

## 3. Results

The results were obtained by comparing 450 flies (15 experiments × 30 flies/experiment) per group, with each group consisting of 15 subgroups of 30 flies of mixed sex (15 of each sex). In total, 15 experiments per group were conducted, contributing to the reported results. The total distance traveled and velocity data were calculated as mean value and standard deviation for all 450 flies in the group. Due to the large number of interactions, SIN measures were presented as the mean and standard deviation for each group, and calculated by performing a temporal network analysis in which each minute of the experiment is represented by a separate network snapshot [26]. Measurements were then taken across each representation of networks, which were summarized in the distribution for each group.

In other words: (i) the mean values for all SIN measures were calculated for each one-minute snapshot of video recordings; (ii) the mean values from these one-minute snapshots were used to calculate the mean for each video recording; and (iii) finally, the mean values of SIN measures of the 15 video recordings per group were computed to derive the overall mean for each group *(dFMR1^B55^* or *w^1118^*). The results of the activity and SNA are summarized and presented in Figure 2. 

### 3.1. Activity Analysis 

*dFMR1^B55^* flies, compared to the control *w^1118^* group, showed statistically significant decreased activity levels, manifested by lower total distance traveled (Figure 2a; 17,436.00 ± 1688.00 mm vs. 30,642.00 ± 2408.00 mm; *p* < 0.0001, t = 4.54) and lower average velocity (Figure 2b; 4.65 ± 0.45 vs. 8.17 ± 0.64; *p* < 0.0001, t = 4.54).

### 3.2. Social Network Analysis (SNA) 

*Total edges.* The number of interactions (total edges) was statistically significantly lower in *dFMR1^B55^* mutants (Figure 2c; 339.20 ± 4.57 vs. 231.00 ± 6.42; *p* < 0.0001, t = 13.57).

*Average degree.* Likewise, the average number of interactions that the fly has participated in, presented as average degree, was statistically significantly lower in the *dFMR1^B55^* than in the *w^1118^* group (Figure 2d; 15.95 ± 0.44 vs. 22.75 ± 0.30; *p* < 0.0001, t = 12.67). 

*Average strength.* Analysis of average in-strength and average out-strength by number of interactions has demonstrated that the *dFMR1^B55^* group had initiated and received statistically significantly fewer interactions (Figure 2e; 16.42 ± 0.75 vs. 19.23 ± 0.31; *p* < 0.0001, t = 3.40; both). However, analysis of average in-strength and average out-strength by duration revealed statistically significantly higher values in the *dFMR1^B55^* group (Figure 2f; average in-strength: 3252.00 ± 181.00 vs. 1663.00 ± 43.79; *p* < 0.0001, t = 8.29; both).

*Network density.* Network density was statistically significantly reduced in *dFMR1^B55^* mutants compared to *w^1118^* flies (Figure 2g; 0.27 ± 0.01 vs. 0.40 ± 0.01; *p* < 0.0001, t = 11.47).

*Global efficiency*. The global efficiency of *dFMR1^B55^* was statistically significantly lower compared to the *w^1118^* networks (Figure 2h; 0.61 ± 0.01 vs. 0.71 ± 0.00; *p* < 0.0001, t = 12.85).

*Heterogeneity*. There was a statistically significantly higher degree of heterogeneity in the *dFMR1^B55^* group relative to the *w^1118^* fly groups (Figure 2i; 0.44 ± 0.01 vs. 0.29 ± 0.01; *p* < 0.0001, t = 16.80).

*Assortativity*. Additionally, assortativity was statistically significantly higher in the *dFMR1^B55^* groups than in the *w^1118^* groups (Figure 2j; 0.06 ± 0.01 vs. −0.02 ± 0.01; *p* < 0.0001, t = 6.87).

*Transitivity.* Transitivity did not show statistically significant differences between the *dFMR1^B55^* and *w^1118^* groups (Figure 2k; 0.47 ± 0.01 vs. 0.47 ± 0.01; *p* = 0.95, t = 0.46).

*Average clustering coefficient.* Similarly, the clustering coefficient showed no statistically significant difference between the two tested groups (Figure 2l; 0.47 ± 0.01 vs. 0.48 ± 0.00; *p* = 0.81, t = 0.24). However, analysis of the clustering coefficient weighted for the duration of interactions showed that *dFMR1^B55^* flies have a higher clustering coefficient than *w^1118^* (0.06 ± 0.00 vs. 0.05 ± 0.00; *p* < 0.0001, t = 4.04).

*Average betweenness centrality. dFMR1^B55^* flies showed statistically significantly higher betweenness centrality than *w^1118^* (Figure 2m; 0.04 ± 0.00 vs. 0.02 ± 0.00; *p* < 0.0001, t = 0.24). Similar results were obtained when weights for the count and duration were applied (weight = count: 0.05 ± 0.00 vs. 0.03 ± 0.00; *p* < 0.0001, t = 20.22; weight = duration: 0.06 ± 0.00 vs. 0.04 ± 0.00; *p* < 0.0001, t = 4.04). 

*Average closseness centrality.* Average closeness centrality was statistically significantly lower for SINs in the *dFMR1^B55^* groups than in the *w^1118^* groups (Figure 2n; 0.56 ± 0.00 vs. 0.61 ± 0.00; *p* < 0.0001, t = 10.31). Similar results were obtained when weight for count or duration were applied (weight = count: 0.48 ± 0.00 vs. 0.52 ± 0.00; *p* < 0.0001, t = 9.17; weight = duration: 0.01 ± 0.00 vs. 0.01 ± 0.00; *p* < 0.0001, t = 18.44).

*Modularity.* Modularity was statistically significantly higher in the *dFMR1^B55^* than in *w^1118^* groups (Figure 2o; 0.21 ± 0.00 vs. 0.17 ± 0.00; *p* < 0.0001, t = 8.55). Similar results were obtained when weight for count or duration were applied (weight = count: 0.24 ± 0.00 vs. 0.19 ± 0.00; *p* < 0.0001, t = 8.09; weight = duration: 0.26 ± 0.01 vs. 0.21 ± 0.01; *p* < 0.0001, t = 5.92).

### 3.3. Localization of Social Interactions 

Using heatmaps, we were able to visualize the location of interactions better. Both groups of flies show a preference for the arena boundaries, a phenomenon known as thigmotaxis [29]. Although we did not perform additional statistical analysis, it appears that *dFMR1^B55^* have a lower number of interactions and they tended to aggregate closer to the arena edges (Figure 3).

## 4. Discussion

The current study is among the first studies to describe impaired SI in the FXS model of *D. melanogaster*. According to our results, *dFMR1*^B55^ flies exhibited hypo-activity and fewer connections within their networks. Additionally, they demonstrated a reduced ability to efficiently transmit information due to fewer alternative pathways for information transmission, a higher variability in the number of interactions they achieved among themselves and the fact they tended to stay near the boundaries of the testing chamber. Despite participating in fewer interactions, *dFMR1*^B55^ flies tended to spend more time engaged in each interaction. This observation was based on the statistically significant: (i) lower parameters which are linked to number of interactions (average in-strength and average out-strength by number of interactions) in *dFMR1*^B55^ flies compared to *w^1118^* controls; and (ii) higher parameters which are linked to duration of the interaction (average in-strength and average out-strength by duration) for *dFMR1*^B55^ flies compared to *w^1118^* controls. However, they primarily interacted with individuals who had a similar number of interactions. In addition, the distances between them were longer and the spread of information was slower. Interestingly, our results suggested that there were individual *dFMR1*^B55^ flies in the network that played important roles as intermediaries connecting different parts of the network. Higher modularity suggests that a *dFMR1*^B55^ network can be divided into distinct communities with more connections within the community than outside of it. Conversely, two groups of flies (wild-type and *dFMR1*^B55^) exhibited similar local connectivity patterns. Briefly, *dFMR1*^B55^ flies achieved a lower total and average number of SIs, and exhibited alterations in various SIN measures compared to wild-type flies. These alterations suggest mobility, connectivity, and overall network organization changes in *dFMR1*^B55^ flies.

As described by Svetec and Ferveur (2005), [30] if social experience measured among male flies only, in the absence of females and food, males displayed homosexual courtship and aggressive behaviors, the frequency, intensity and directionality of which varied according to their experience. In addition, as reviewed in Jezovit et al. (2021) [31], experiments with flies in a homogeneous group revealed a ‘touching’ behavior, where the foreleg of one fly (the ‘interactor’) makes contact with the arista, head, body, wing, or leg of another fly (the ‘interactee’). Wice and Saltz (2021) described how females have different networks than males. Based on previous findings, here, we attempted to study both sexes mixed into the same groups, and to analyze the general SI of the group, regardless of courtship. Future experiments could analyze the difference between SI of males and females, as well as the influence of courtship on SI in the FXS model of *Drosophila melanogaster* [24]. 

In contrast to previous studies that researched SI in *dFMR1* mutants, all presented results obtained in the current study were based on SNA as a powerful statistical tool. SNA has been used in the last 20 years to analyze collective animal behaviors and identify group SI patterns (reviewed in: [31]). There has also been a growing interest in utilizing SNA to examine the social behavior of *D. melanogaster* [16,18,22,24,26,31,32,33,34]. In addition, our study, for the first time, provides results of SI in the FXS model of fruit flies obtained using a combination of novel tools: the Drosophila Shallow Chamber and the open-source Python data processing pipeline for analysis of SI in *D. melanogaster*. Specifically, a validation of SNA in research with fruit flies, as a method chosen and used in the current study, was recently published [25].

Only a few studies have focused on SI in the *D. melanogaster* model of FXS. This model is characterized by absence of dFMRP, and its phenotype is directly associated with dFMRP deficiency. According to previous molecular research, SI impairment in the *D. melanogaster* model of FXS is based on lack of dFMRP. (reviewed in: [10,11]). Dockendorf et al. (2002) showed that *dFMR1^B55^* mutants exhibit altered courtship and mating behavior [13]. Male flies failed to advance to more intricate phases of courtship beyond following and tapping, resulting in shorter time spent in courtship activities [13,35]. These findings may resemble the loss of interest in engaging in SI frequently present in humans with FXS [13,36]. In addition, Bolduc et al. (2010) used a different methodology to study SI in *dFMR1^B55^* and *dFMR1^3^* mutants [37]. Specifically, they used two chambers separated by a plastic mesh to research parameters related to SI and demonstrated that both *dFMR1* mutants and control groups tended to stay near the boundaries of the testing chamber. This behavior, known as thigmotaxis, has been well-documented and observed in fruit flies [37,38]. Our results are in accordance with previous observations and confirmed the presence of thigmotaxis in *dFMR1^B55^* using a novel Shallow Chamber. The Shallow Chamber was developed by Maze Engineers to study groups of flies using a design to prevent the flies from obscuring one another. In addition, Bolduc et al. (2010) showed hypoactivity in both *dFMR1* mutants, which is consistent with our data [37]. Hypoactivity was also previously found in *dFMR1* larvae [39]. Furthermore, the likelihood of SI, measured by the interfly distance, was lower in *dFMR1^B55^* than in wild-type flies [37]. This is in line with our findings: *dFMR1^B55^* flies had fewer interactions with greater interfly distance, as shown by lower closeness centrality. Moreover, it was shown that *dFMR1* mutants and wild-type flies display different spatial distributions within the chamber. Wild-type flies were uniformly distributed, while *dFMR1^B55^* preferred the chamber interior [37]. Interestingly, *dFMR1^3^* mutants showed a phenotype with shared characteristics of both *dFMR1^B55^* and wild-type flies. Our results also indicate a uniform distribution of wild-type flies in the social network. Lower betweenness centrality and lower modularity primarily mean that each fly is equally essential in the information transmission process and that wild-type SINs are more uniform than *dFMR1^B55^* SINs. Additionally, researchers previously noticed that *dFMR1* mutants made frequent, irregular stops. Such a behavior was described as an arrhythmic phenotype by Dockendorff and colleagues in 2002 and Bolduc and colleagues in 2010 [13,37]. This behavior, described as a basic form of dyspraxia, agrees with a decreased receptive response observed in *dFMR1^B55^* flies [13,37]. While the SNA used in our study does not provide information regarding the regularity and duration of individual stops out of SI, other results in our study are consistent with limited data on SI in the FXS model of *D. melanogaster*. However, our results provide more data about their mutual interactions, and their role in the social network over time.

The importance of the current research on SI in the *Drosophila* FXS model is based on the fact that the social network structure is highly influenced by genotype [16,24,32,33]. Different *D. melanogaster* strains exhibit differences in SINs, indicating the potential influence of genes on SIN structure [32]. Wice and Saltz (2021) analyzed five commonly studied SIN measures (in-strength, out-strength, clustering coefficient, and betweenness centrality) in 40 randomly chosen inbred lines of flies [24]. They confirmed that an individual’s genotype was a significant indicator for all network measures examined [24]. The authors calculated broad-sense heritability, a genetic parameter used to estimate the proportion of phenotypic variation due to genetic factors [40]. Betweenness centrality displayed the highest broad-sense heritability, with genotype accounting for about 17% of the variation in this network measure [24]. Additionally, Wice and Saltz (2023) demonstrated that SIN measures depend on the individual’s genotype and the genotypes of other individuals within the network [24,33]. Furthermore, Alwash et al. (2021) investigated the characteristics of SI in flies mutated in for, a pleiotropic gene regulating several metabolic, physiological and behavioral phenotypes [32]. They demonstrated that the positions within the group are inherited and that the flies that form SINs are robust over time [32]. These SINs are characteristic strain-dependent social networks and are known as group phenotype [32]. Here, we describe in detail that the genotype in our focus (*dFMR1* mutants) has made a significant contribution to the investigated parameters of SI and suggest that *dFMR1* models could be used in various biomedical and pharmacological studies based on SI impairment.

Aside from genetic factors, variations in SINs could be due to other factors, such as social experience. Jezovitz et al. (2021) compared the results of studies that focused on social networks in *D. melanogaster*, and observed that, despite methodological differences, studies agree that isolated flies exhibit distinctly altered SINs [31]. Isolated flies form SINs with increased global efficiency and lower betweenness centrality and these characteristics are recorded regardless of fly age [16,18,22,31]. Moreover, significant variability across all SIN measures in isolated flies could suggest that a lack of social experience results in less predictable networks [31]. Although, in the current study, *dFMR1* mutants were not isolated, variability of a few SIN measures was also observed in these mutants. These findings suggested that other factors influence variability of some SIN measures in addition to isolation. Thus, further behavioral and molecular studies are needed to identify more details in SI impairments in *dFMR1* mutants, as an excellent model for pharmacological screening studies.

## 5. Conclusions

A combination of Drosophila Shallow Chamber and SNA is a valuable method for SI research in fruit flies. Using this method, we demonstrated that *dFMR1*^B55^ mutants are characterized by SI impairment and established a group phenotype of this model. These findings could enable a better understanding of SI in *dFMR1*^B55^ and the potential development of pharmacological research and rapid pharmacological screening in the field of FXS. There is no approved targeted treatment for FXS, and the validation of the *dFMR1^B55^* model would contribute to targeted treatment development in the field of FXS. 

## Figures and Tables

**Figure 1 biology-13-00432-f001:**
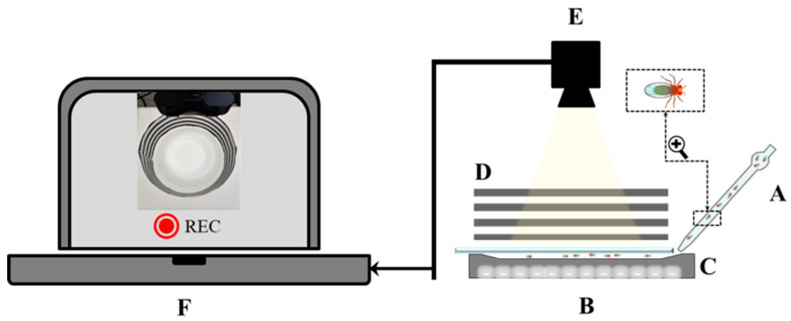
Schematic representation of the experimental design: A—the flies were transferred by the aspiration to the B—Drosophila Shallow Chamber, a specially designed shallow space for creating a monolayer of flies forcing them into social interaction; C—12 × 12 inch fluorescent light array of 850 nm LEDs; D—translucent checkered black and white paper surrounding the Chamber to stimulate the movement of flies; E—camera for video recording, F—computer for storing and analyzing recordings.

**Figure 2 biology-13-00432-f002:**
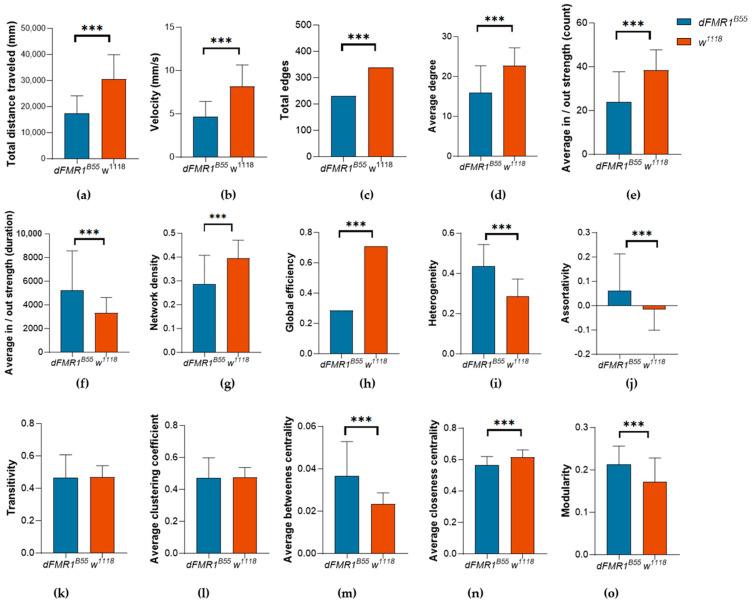
Differences between *dFMR1^B55^* and *w^1118^* in activity and Social Interaction Network (SIN) measures. Column bar graph (mean and standard deviation) for 15 *dFMR1^B55^* (*n* = 450 flies) and 15 *w^1118^* groups (*n* = 450 flies), for the following measures: (**a**) total distance traveled expressed in mm; (**b**) velocity expressed in mm/s; (**c**) total edges; (**d**) average degree; (**e**) average in/out strength weighted for count; (**f**) average in/out strength weighted for duration; (**g**) network density; (**h**) global efficiency; (**i**) heterogeneity; (**j**) assortativity; (**k**) transitivity; (**l**) average clustering coefficient; (**m**) average betweenness centrality; (**n**) average closeness centrality; and (**o**) modularity. Data are extracted from 15 min videos using FlyTracker and analyzed using the Python data processing pipeline. *p*-values less than 0.05 are taken as significant. Column bars represent the mean values of 450 flies per group (15 experiments with 30 flies per group) with the whiskers representing the standard deviation. This information can be useful for understanding the high variability in almost all the graphs. Abbreviations: *** *p* < 0.0001; *dFMR1^B55^*-*Drosophila melanogaster* model of fragile X syndrome; *w^1118^*—wild type, *Drosophila melanogaster.*

**Figure 3 biology-13-00432-f003:**
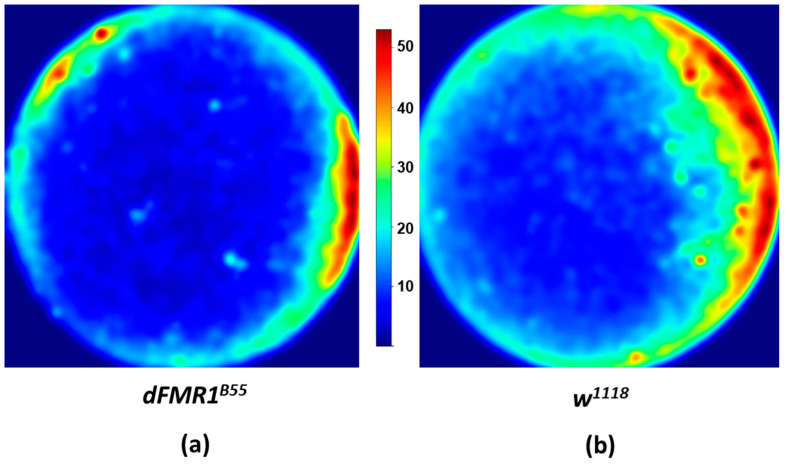
*dFMR1^B55^* and *w^1118^* heatmaps of localization of Social Interactions (SIs). Visualization shows that both groups prefer for the arena boundaries, and it appears that the *dFMR1^B55^* groups have fewer interactions than *w^1118^*. The visualization was made for groups in total for the *dFMR1^B55^* group (**a**) and *w^1118^* group (**b**). Abbreviations: *dFMR1^B55^*-*Drosophila melanogaster* model of fragile X syndrome; *w^1118^*—*Drosophila melanogaster* wild-type. From blue color to red color: from infrequent to frequent Sis.

## Data Availability

Data will be made available upon request.

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
