# Peer review of "Using a Combination of Novel Research Tools to Understand Social Interaction in the Drosophila melanogaster Model for Fragile X Syndrome"

_biology, 2024, doi:10.3390/biology13060432_

Round 1

Reviewer 1 Report

Comments and Suggestions for Authors

The research is focused on investigating the social interactions (SI) in FXS model of Drosophila employing the Drosophila Shallow Chamber, developed by the company Maze Engineers, and processing the data through the Python-based Social Network Analysis (SNA) pipeline.

FXS is a developmental disorder and one of the major monogenic mutations-dependent disorders leading to intellectual disability and autism. It depends on a mutation in the fmr1 gene causing an increase of >200 CGG trinucleotide repeats. The Drosophila model used here is a fmr1 null fly, previously employed in literature as cited, generated by excision of 2.5kb of the gene, rendering the protein product inactive.

15 experimental and 15 control groups of 30 flies each were recorded in the Drosophila Shallow Chambers for 15 minutes, and videos were processed by the SNA pipeline. The SNA pipeline takes account of two coordinates, the node, representing each individual fly, and the edge, representing the links between flies. The weight of the link between each fly and another is considered in terms of the number and duration of interactions. The researchers also established three criteria for determining a social interaction: the distance between flies being within two and a half body lengths, the angle between flies being less than 160 degrees, and minimum duration of interaction being over 0.6 seconds.

The researchers’ analysis revealed decreased SI in fmr1 null flies compared to w1118 control flies.

Minor points:

1-The protocol doesn’t clearly describe whether male and female flies were placed in the Drosophila Shallow Chambers together and whether the researchers accounted for any sex differences during the analysis. This raises questions on whether the SI analyzed includes courtship interactions and how the data would differ when carrying the same analysis with just flies of the same sex. However, both experimental and control groups were analyzed using the same protocol, so this does not raise any doubts regarding the conclusions drawn from the study but only a necessity for a deeper analysis that accounts for sex differences and social interactions independent of courtship.  

2-The description of the Drosophila Shallow Chamber would benefit from a supplemental movie.

3-The authors do not offer an explanation regarding the statistically significant higher average in-strength and average out-strength by duration data obtained compared to the lower average in-strength and average out-strength by number of interactions for fmr1 null flies compared to w1118 controls.

Overall, the study outlined in this paper rests on a robust experimental design and yields conclusive results. It provides clear and strong evidence of SI deficits in an FXS Drosophila model using a well-defined protocol for analyzing sociability in flies. I'll consider its publication after the authors resolve the minor points.

Author Response

Dear Reviewer 1,

Thank you for your comments regarding our manuscript ID biology-3002745. As you can see below, we have addressed all comments and suggestions, and have improved our manuscript accordingly. Please note that we have also enhanced the English throughout the entire manuscript.

Thank you,

Dr. Dragana Protic, corresponding author

  1. Comment: The protocol doesn’t clearly describe whether male and female flies were placed in the Drosophila Shallow Chambers together and whether the researchers accounted for any sex differences during the analysis. This raises questions on whether the SI analyzed includes courtship interactions and how the data would differ when carrying the same analysis with just flies of the same sex. However, both experimental and control groups were analyzed using the same protocol, so this does not raise any doubts regarding the conclusions drawn from the study but only a necessity for a deeper analysis that accounts for sex differences and social interactions independent of courtship.

Answer: Thank you for your observation and suggestion. We accept your comments and modified our article accordingly.  Please see Abstract: line 49, Methods/2.3 Experimental Design: pg 4, lines 135-137, and Discussion pg 10, lines 367-378.

  1. Comment: The description of the Drosophila Shallow Chamber would benefit from a supplemental movie

Answer: A supplemental video was added within our article, and it is mentioned in Methods. Please see lines 144-145.

  1. Comment: The authors do not offer an explanation regarding the statistically significant higher average in-strength and average out-strength by duration data obtained compared to the lower average in-strength and average out-strength by number of interactions for fmr1 null flies compared to w1118 controls.

Answer: Thank you for your suggestion. We accepted your comments and add explanation at the beginning of the Discussion. Please see lines 346-350.

Reviewer 2 Report

Comments and Suggestions for Authors

The paper is interesting even though some changes are needed:

- the last part of the graphical abstract (red panels) should be improved comparing similar phenotypes between the two groups

- a scheme of the swallow chamber describin all the steps of the experiment will be very useful to the reader

- I suggest to precisely indicate how many experiments have been done which contribute to the "mean" reported in results described in paragraph 3 of the paper?  This information will be useful also for understanding the high variability in almost all the graphs reported in Figure 1

- I suggest to increase the dimension of the Figure 1 to easily read all the written part

- I propose to the authors to stronger support the relation between the altered SNAs of dFmr1 mutants and altered molecular mechanisms in which FMRP protein is involved, which were never mentioned in the paper

- References 21 and 23 are the same

Comments on the Quality of English Language

English should be revised

Author Response

Dear Reviewer 2

Thank you for your comments regarding our manuscript ID biology-3002745. As you can see below, we have addressed all comments and suggestions, and have improved our manuscript accordingly. Please note that we have also enhanced the English throughout the entire manuscript.

Thank you,

Dr. Dragana Protic, corresponding author

  1. Comment: the last part of the graphical abstract (red panels) should be improved comparing similar phenotypes between the two groups.

Answer: Thank you for your comments. Please find a new version of graphical abstract which included changes in right red panel according to your suggestion.

  1. A scheme of the swallow chamber describin all the steps of the experiment will be very useful to the reader.

Answer: Thank you for your comments. Please find a scheme (a new Figure 1) that described all the steps of the experiment.

  1. I suggest to precisely indicate how many experiments have been done which contribute to the "mean" reported in results described in paragraph 3 of the paper? This information will be useful also for understanding the high variability in almost all the graphs reported in Figure 1.

Answer: Thank you for your comments. We precisely indicated how many experiments have been done which contribute to the "mean" reported in results described in paragraph 3 of the paper. Please find lines 233-236 and 260-263.

  1. I suggest to increase the dimension of the Figure 1 to easily read all the written part.

Answer: The dimension of the Figure 1 is increased.

  1. I propose to the authors to stronger support the relation between the altered SNAs of dFmr1 mutants and altered molecular mechanisms in which FMRP protein is involved, which were never mentioned in the paper.

Answer: Thank you for your observation. I strongly agree with you and added a few sentences about dFMRP roles. Please see lines: 76-79; 94-98 (Introduction) and 389-392 (Discussion).

  1. References 21 and 23 are the same.

Answer: Thank you for your observation. The same reference is excluded in this version.

Round 2

Reviewer 2 Report

Comments and Suggestions for Authors

The new version of the manuscript is fine.